# Risk of revision arthroplasty surgery after exposure to physically demanding occupational or leisure activities: A systematic review

**Elena Zaballa**[1,2]*, **E. Clare Harris**[1,2], **Cyrus Cooper**[1], **Catherine H. Linaker**[1,2], **Karen Walker-Bone**[1,2]

1 Medical Research Council Life Course Epidemiology Centre, University of Southampton, Southampton, United Kingdom, 2 Medical Research Council Versus Arthritis Centre for Musculoskeletal Health and Work, University of Southampton, Southampton, United Kingdom

* ez@mrc.soton.ac.uk

**Funding:** KWB and CC were co-applicants on the MRC Versus Arthritis (formerly Arthritis Research

## Abstract

### Introduction

Lower limb arthroplasty is successful at relieving symptoms associated with joint failure. However, physically-demanding activities can cause primary osteoarthritis and accordingly such exposure post-operatively might increase the risk of prosthetic failure. Therefore, we systematically reviewed the literature to investigate whether there was any evidence of increased risk of revision arthroplasty after exposure to intensive, physically-demanding activities at work or during leisure-time.

### Methods

We searched Medline, Embase and Scopus databases (1985—July 2021) for original studies including primary lower limb arthroplasty recipients that gathered information on physically-demanding occupational and/or leisure activities and rates of revision arthroplasty. Methodological assessment was performed independently by two assessors using SIGN, AQUILA and STROBE. The protocol was registered in PROSPERO [CRD42017067728].

### Results

Thirteen eligible studies were identified: 9 (4,432 participants) after hip arthroplasty and 4 (7,137participants) after knee arthroplasty. Narrative synthesis was performed due to considerable heterogeneity in quantifying exposures. We found limited evidence that post-operative activities (work or leisure) did not increase the risk of knee revision and could even be protective. We found insufficient high-quality evidence to indicate that exposure to physically-demanding occupations increased the risk of hip revision although "heavy work", agricultural work and, in women, health services work, may be implicated. We found conflicting evidence about risk of revision hip arthroplasty associated with either leisure-time or total physical activities (occupational or leisure-time).

UK) Centre for Musculoskeletal Health and Work award (ref 22090) (https://www.versusarthritis.org/). This award provided support for the PhD studentship of EZ and support for the salaries of CHL and ECH. The funders had no role in the study design, data collection and analysis, decision to publish, or preparation of the manuscript.

**Competing interests:** The authors have declared that no competing interests exist.

## Conclusion

There is currently a limited evidence base to address this important question. There is weak evidence that the risk of revision hip arthroplasty may be increased by exposure to physically-demanding occupational activities but insufficient evidence about the impact on knee revision and about exposure to leisure-time activities after both procedures. More evidence is urgently needed to advise lower limb arthroplasty recipients, particularly people expecting to return to jobs in some sectors (e.g., construction, agriculture, military).

## Introduction

Hip and knee replacements have been routinely indicated for the treatment of end-stage arthritis over the past 40 years [1, 2]. The demand for these operations is increasing both because of the ageing population but also because of growth in rates of surgery amongst people aged < 60 years. According to data from the National Joint Registry, the number of primary hip and knee replacements performed in England, Northern Ireland and Wales amongst people aged below 60 years increased by 25% and 20% respectively from 2010 to 2018 [3]. Future projections point towards an even greater increase by 2030 and 2035 [4–6].

Although highly effective interventions [7, 8], hip and knee replacements may fail over time necessitating revision surgery to the replaced joint. Revision surgery is more complex than primary arthroplasty with poorer outcomes [9] and a greater economic burden on health services [10, 11]. Survival rates after arthroplasty are lower amongst younger recipients. One studied reported higher failure rate in hip arthroplasty recipients aged <60 years [12]. Another study reported that, compared with the 15% lifetime risk of revision amongst those aged 60 years, rates of hip revision were 29.6% and of knee revision were 35.0% amongst those aged 50–54 years [13]. These age differences are at least partly explained by sex (greater risk among male recipients) but also by different indications for primary surgery, type of prosthesis and fixation method [14] but there is need for a better understanding of the impact of other factors on implant survival.

Modern arthroplasty techniques derive from the 1960s (hip) and 1970s (knee). Since then, there have been vast improvements in component materials, geometry and fixation as well as surgical techniques, leading to shorter length of hospital stay, more conservative surgery and better outcomes. Concerned about the consequences of damage to the prostheses, surgeons in the past generally urged caution to patients about their participation in sport and LTPA. Moreover, lower limb arthroplasty surgery was typically offered relatively late in the course of joint failure and thus the majority of patients were elderly and not expecting to return to the labour market. Despite the limited evidence against engaging in LTPA post arthroplasty, the consensus amongst orthopaedic surgeons has been to advise caution [15].

There is considerable evidence that exposure to physically-demanding work which mechanically loads the hip (e.g. heavy lifting) or knee (e.g. kneeling), increases the risk of primary osteoarthritis at those sites [16–19]. People aged <60 years at the time of their arthroplasty are likely to need to return to their occupation and possibly engage in other physically-demanding activities during leisure-time. A previous systematic review evaluated the evidence that host factors were associated with aseptic loosening after arthroplasty [20]. They identified three studies involving 178 hip arthroplasties which reported "activity levels" and an increased risk of aseptic loosening after total hip arthroplasty (THA). Unfortunately, this review did not separate activities occurring at work from those occurring outside the workplace.

Furthermore, although aseptic loosening is a common mechanism necessitating revision surgery, it only accounts for 25% of hip revision arthroplasties [21] and 20% of knee revision operations [22]. Therefore, to fill this gap, we undertook a systematic review of the published literature in order to explore the evidence about the risk of revision arthroplasty surgery related to physically-demanding activities performed (a) at work and (b) during leisure-time.

## Material and methods

A protocol of the systematic review was registered in PROSPERO (registration number CRD42017067728). Following the Population, Intervention, Comparison and Outcome (PICO) format, our research question was as follows: amongst adults aged over 18 years at the time of primary hip or knee arthroplasty, undertaken for any common indication, what was the effect of exposure to physically-demanding activities (a) at work and (b) in leisure on the risk of revision surgery performed for any reason other than for reasons of infection.

### Search strategy

Our search was conducted in three electronic databases: MEDLINE and Embase using the Ovid search engine, and in Scopus (S1 File), limited to studies published in peer-reviewed journals, from January 1985 to week 5 June 2021 (in Medline), and 7 July 2021 (in Embase and Scopus), in English or Spanish languages. Duplicates were removed, and letters, notes, editorials and editorial commentaries were also excluded. However, when a conference abstract was found, we checked whether a full paper was subsequently published. In addition, reference lists from all full papers retrieved, as well as the systematic reviews found during the search, were checked to find any additional relevant studies not covered by the MeSH terms or key words used in the search.

### Inclusion and exclusion criteria

To be eligible for inclusion, publications were randomised controlled trials, case-control or cohort studies including adults with primary hip or knee arthroplasty, followed-up for more than 12 months post-operatively, and in whom information was collected about either, or both, physically-demanding occupational or leisure-time activities and in which rates of revision arthroplasty were recorded. We excluded those studies that investigated: i) patients with only inflammatory arthritis or other specific rarer pathologies (e.g., haemophilia); ii) hip or knee surgical procedures other than total replacement and joints other than hip or knee; iii) risk factors related to operative procedure or nature of prosthesis only (e.g., surgical approach); and iv) non-elective arthroplasties. Studies were also excluded if: participants were under 18 years of age at the time of the arthroplasty; the indication for revision arthroplasty was exclusively infection; or the outcome measured was not revision surgery (e.g. volume of polyethylene wear).

### Screening

Screening of titles and abstracts was initially undertaken by one reviewer (EZ) who classified papers as "eligible", "ineligible" or "uncertain whether eligible or not" for inclusion in the review. A second reviewer (ECH and CHL), checked all papers classified as uncertain to be suitable for inclusion (n = 229) and where consensus was not reached, discussed with a third reviewer (KWB). Additionally, a random sample of 10% of those deemed by the first reviewer as "eligible" or "ineligible" were also screened by a second reviewer but it was demonstrated that none of these papers had been misclassified. Once full text papers had been agreed and

selected, two reviewers (EZ, ECH or CHL) independently reviewed the full texts for suitability for inclusion. Discrepancies were discussed by both reviewers and, if consensus was not reached, with a third reviewer (KWB).

## Data extraction

Data were extracted from included articles independently by two reviewers (EZ and ECH / CHL) according to a pre-defined proforma. Data extraction included: author and year of publication, study design, country, site of procedure, duration of follow-up, indication for primary arthroplasty, sample size, age at the time of primary operation and age at revision (if provided), number lost to follow-up, operation-related factors, definition of revision, type of physical activity (undertaken at work and/or during leisure time), method of measurement of physical activity including how exposure to physical activity that loads the joint was categorised (e.g. "active vs inactive" or "high, medium, low"), covariates considered, risk estimates and source of funding where available. Findings from the data extracted were reported according to exposure to physical activity: i) occupation and occupational activities, ii) leisure-time physical activities (LTPA), and iii) total physical activity.

## Quality assessment

To evaluate the methodological quality to address our specific research question (S1 and S2 Tables), we used a modified version of the Scottish Intercollegiate Guidelines Network (SIGN) checklist for observational studies [23] alongside the Assessment of Quality in Lower Limb Arthroplasty (AQUILA) checklist (which was specifically developed to assess quality of lower limb arthroplasty studies) [24] for cohort studies and the Strengthening the Reporting of Observational studies in Epidemiology (STROBE) [25] checklist for case-control studies. Two reviewers (KWB, EZ) independently assessed each study, and subsequently compared their ratings, discussing any discrepancies until consensus was reached about any potential bias and the direction of its effect.

## Results

In total, 20,274 citations were identified. Only three further citations, published prior to 1985, were retrieved by hand searching bibliographies of relevant papers and systematic reviews. After removing duplicates, 11,307 titles and abstracts were screened, yielding 50 studies that were potentially relevant, for which full texts were obtained (Fig 1). Assessment of the full text publications resulted in the exclusion of a further 37 studies, leaving 13 papers eligible for inclusion in this review.

Table 1 summarises the main characteristics of the 13 studies retrieved. Published between January 1983 and July 2021, nine related to risk of revision after primary THA [26–34], and four the risk of revision after primary total knee arthroplasty (TKA) [35–38]. In terms of study design, there were ten longitudinal studies; two prospective [29, 30] and eight retrospective studies [26, 27, 31–33, 35, 37, 38], and three case-control studies [28, 34, 36]. The main reasons for scoring poorly on quality assessment were: a lack of detail regarding how the activity exposures were measured; insufficient information about how the participants were classified into groups exposed to more or less demanding physical activities; insufficient information about selection criteria; and failure to adjust for potential confounders in the analyses. One study reported exposure to LTPA more precisely than was the case for exposure to physically-demanding occupational activities (not stated how many people actually returned to the occupations post-operatively) and therefore, according to our quality assessment criteria, needed to be scored differently for the purposes of this review [28]: it was graded acceptable quality for

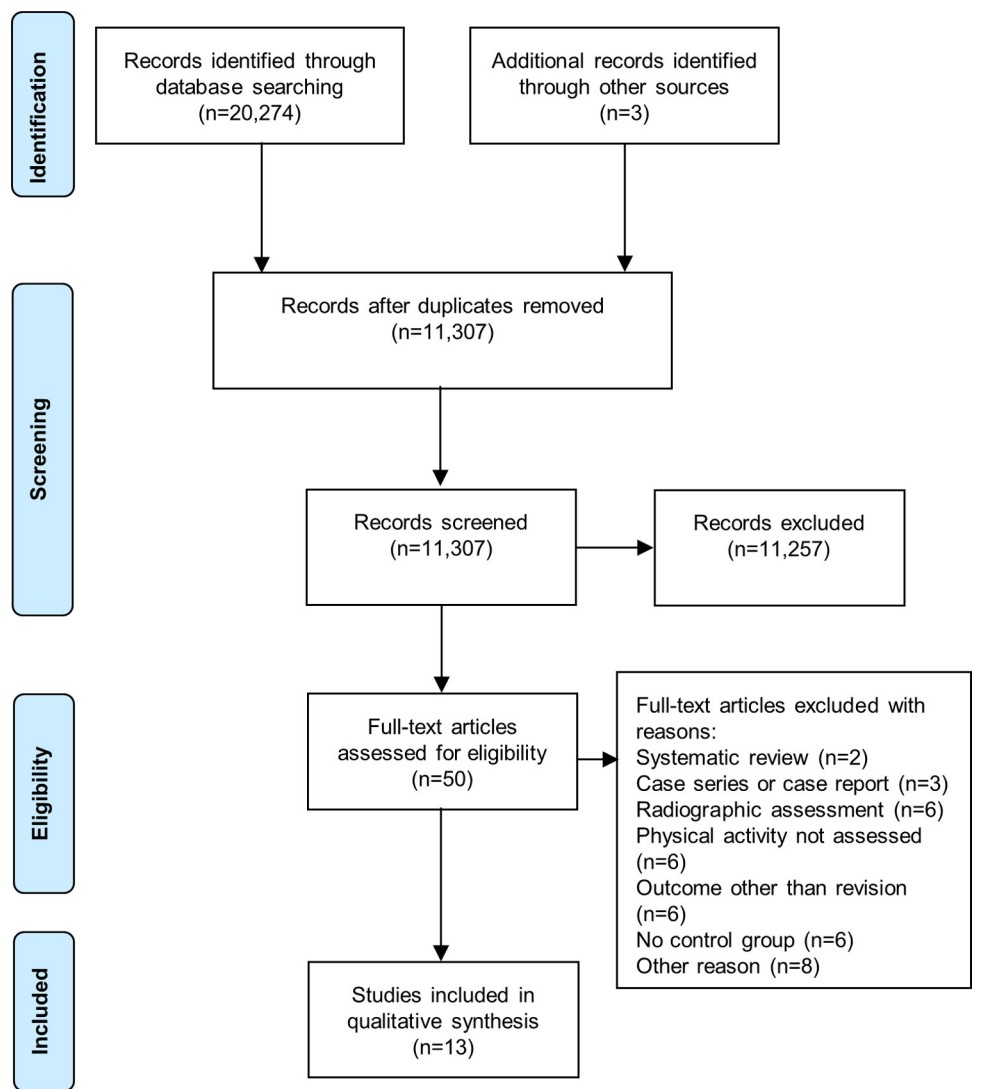

**Fig 1. PRISMA flow diagram for the identification of the studies included.**

LTPA but poor quality for occupation. The quality scoring for the remaining papers were as follows: two were rated as "high quality", six "acceptable", one "poor" and three "very poor".

The number of study participants ranged from 18 [33] to a maximum of 2,016 [37], and the post-operative follow-up from 4.9 [27] to 11 years [33, 38]. The average age of patients at the time of surgery was between 55 and 73 years, with primary osteoarthritis (OA) as the main indication (prevalence >60%) for both THA and TKA. Overall, studies recruited more women than men.

Lower limb arthroplasties were performed either by a single [35], two [27, 33, 38] or more orthopaedic surgeons [29–32, 36, 37]. Unilateral procedures were more frequent, but six studies also included people undergoing bilateral arthroplasty (between 1–50% of participants) [26, 28, 29, 32, 35, 38]. Different types of implant fixation were used: cemented in five studies [26, 27, 29, 30, 38]; uncemented in two studies [28, 33]; hybrid in two studies [32, 36]; and one study included all types of fixation [31].

Definition of the outcome differed between studies. For some authors, revision included all revision TKA procedures [37, 38] recorded in the registries and one of THAs included "a

**Table 1. Description of the eligible studies retrieved by site of primary arthroplasty (hip or knee) and year of publication.**

| Author, Country | Year | Study design | Number of participants | Age (years) at primary arthroplasty | Gender | Indication for arthroplasty | Fixation technique | Duration of follow-up / mean time to revision | Definition of revision | Quality[a] / Risk of bias |
|---|---|---|---|---|---|---|---|---|---|---|
| **TOTAL HIP ARTHROPLASTY** | | | | | | | | | | |
| Dubs et al, Switzerland [26] | 1983 | Retrospective | 110 participants (152 THAs) operated between 1970 and 1980 | Mean: 55.4 (29–68) | All (110) men | Hip OA. Polyarthritis and Bechterew's arthritis patients excluded | Cemented | Mean (range): 5.8 years (1–14) | Revision surgery of the replaced hip joint because of loosening | - / High |
| Kilgus et al, USA [27] | 1991 | Retrospective | 688 patients from the UCLA hip replacement database operated by two surgeons. | Mean: in 25 more physically active patients: 48 / In 663 less physically active patients: 60 | Women:439 Men: 249 | OA (248), avascular necrosis (95), RA and juvenile RA (66) and congenital dysplasia of the hip (44) | Cemented | OA patients More active: mean FU 9.2 years Less active: mean FU 4.9 years Non-OA patients More active: mean FU 10.7 years Less active: mean FU 5.2 years | Hip revision procedure for aseptic loosening | - / High |
| Espehaug et al, Norway [28] | 1997 | CC | 536 cases (primary and revision surgery) and 1,092 controls (primary surgery only) from NAR between 1987 and 1993. Controls matched for gender, age at THA (± 5 years), date of operation (± 30 days) and bilaterality. Response rate: 81% overall (cases and controls) | Median, range: 67 (16–88) | Poorly described in paper "Male patients constituted 43% of the material" | Primary OA: case 67% control 67%; RA: case 3.8%, control 3.6%; Femoral neck fracture: case 9.3%, control 8.9%; Congenital dysplasia: case 11%, control 12% | Cases vs controls: cemented (63% vs 74%), uncemented (28% vs 21%) | | Partial or total revision (exchange or removal of a part or the whole of the hip prosthesis) | + / Low For recreational activity exposure |
| Espehaug et al, Norway [28] | 1997 | CC | 536 cases (primary and revision surgery) and 1,092 controls (primary surgery only) from NAR between 1987 and 1993. Controls matched for gender, age at THA (± 5 years), date of operation (± 30 days) and bilaterality. Response rate: 81% overall (cases and controls) | Median, range: 67 (16–88) | Poorly described in paper "Male patients constituted 43% of the material" | Primary OA: case 67% control 67%; RA: case 3.8%, control 3.6%; Femoral neck fracture: case 9.3%, control 8.9%; Congenital dysplasia: case 11%, control 12% | Cases vs controls: cemented (63% vs 74%), uncemented (28% vs 21%) | | Partial or total revision (exchange or removal of a part or the whole of the hip prosthesis) | 0 / Moderate For occupational exposure |
| Inoue et al, Japan [29] | 1999 | Prospective | 130 (151 THAs) patients performed between October 1978 and August 1988 | Mean (range): 61.5 (32–84) | Women: 111 (130 THA) Men: 19 (21 THA). | OA (103), RA (35) and others (13) | Cemented | Mean (range): 7.5 years, (0.2–15.3) | Failure of the femoral component defined as subsidence of the stem, fracture of the cement or stem or a radiolucent line at the cement-prosthesis interface. Failure of the acetabular component defined as component migration or any new fracture in the cement mantle. | 0 / High |

*(Continued)*

**Table 1.** (Continued)

| Author, Country | Year | Study design | Number of participants | Age (years) at primary arthroplasty | Gender | Indication for arthroplasty | Fixation technique | Duration of follow-up / mean time to revision | Definition of revision | Qualityᵃ / Risk of bias |
|---|---|---|---|---|---|---|---|---|---|---|
| Maurer et al, Switzerland [30] | 2001 | Prospective | 589 primary THAs performed from 1984 to 1993. Participants were categorised into 3 groups according to the type of stem received. 6.8% were lost to follow-up and 184 (31%) died before failure could occur | Mean (± SD): CoCrNi:68.7 ± 9.80 Titanium SS 77: 69.3 ± 9.50 Titanium SLS: 69.5 ± 9.70 | Men (%): CoCrNi: 59 Titanium SS 77: 51 Titanium SLS: 63 | OA diagnosis (%): CoCrNi: 66 Titanium SS 77: 72 Titanium SLS: 68 | Cemented | Median (years): CoCrNi: 10.2 Titanium SS 77: 7.7 Titanium SLS: 5.2 | Revision of the femoral component for aseptic loosening following THA (secondary outcome of the study) | + / Moderate |
| Flugsrud et al, Norway [31] | 2007 | Retrospective | 1,535 patients who underwent THA before January 2001 as recorded on the Norwegian Arthroplasty Register (NAR). Hip replacements performed pre-NAR were identified if the hips were revised after NAR was initiated. 121 people deceased at FU | Mean (± SD):Women: 63 (± 5.8) Men: 63 (± 5.4) | Women: 969 Men: 566 | Primary OA (1,025), dysplasia of the hip (159), hip fracture (147), RA (48) and not recorded (113) | Cemented, uncemented and hybrid (cementless cup and cemented stem) | Not given | Revision due to aseptic loosening of cup, stem, or both | + / Moderate |
| Lübbeke et al, Switzerland [32] | 2011 | Retrospective | 433 patients with complete clinical and radiological data (503 THAs) performed between March 1996—December 1998, and January 2001—May 2003. | Mean (range): 67.7 (30–91) | 58% of the THAs were performed in women | All indications excluding trauma or metastatic disease | Hybrid prosthesis comprising cemented stem and uncemented acetabular component | Mean (range): 94.5 months (50–146) Mean time to revision: 74.8 months, range (57–119) | Focal/linear osteolysis around the femoral component (primary outcome), linear wear of acetabular component, and revision for aseptic loosening in the acetabular or femoral component at 5 and 10 years post-primary THA (secondary outcomes) | + / Low |
| Ollivier et al, France [33] | 2012 | Retrospective | 210 participants identified retrospectively among 843 hip replacements performed by two surgeons between 1995 and 2000. 70 participants who practised high impact sports were matched to 140 people with low activity levels for age at THA (± 5 years), sex, BMI, ASA score, follow-up (± 2 years). | Mean ± SD: 58.76 ± 9.4 in high impact sports group and 58.57 ± 9.2 in low activity group | Men, n (%): 36 (51.4) in high impact activities, and 72 (51.4) in low activities | Charnley Grade A or B, OA, osteonecrosis and developmental dysplasia stage 1 | Uncemented hydroxyapatite (HA) coated stem and uncemented HA-coated titanium alloy acetabular cup | Mean (range): 11 years (10–15) | Revision due to mechanical failure, fracture during athletic activities or radiographic sign of aseptic loosening. Septic loosening cases excluded | + / Moderate |

*(Continued)*

**Table 1.** (Continued)

| Author, Country | Year | Study design | Number of participants | Age (years) at -primary arthroplasty | Gender | Indication for arthroplasty | Fixation technique | Duration of follow-up / mean time to revision | Definition of revision | Quality[a] / Risk of bias |
|---|---|---|---|---|---|---|---|---|---|---|
| Delfin et al, Sweden [34] | 2017 | CC | 27 cases and controls individually matched for sex, age and time since THA (± 2 years) were identified between 2012 and 2014 from the same hospital. Response rate: 90% in cases 73% in controls | Mean ± SD age at THA: 58.7 ± 7.6 in cases and 59.9 ±7.3 in controls | Cases and controls: Women: 17 Men: 10 | Primary OA in 23 cases and 19 controls Secondary OA in 2 cases and 7 controls Unknown in 2 cases and 1 control | Most of the prostheses cemented | Mean ± SD: 11.9 ± 5.2 years for cases and 12.6 ± 5.3 years for controls | Stem and/or cup revised between July 2012 and July 2014 due to loosening or dislocation of prosthesis | ++/ Low |
| **TOTAL KNEE ARTHROPLASTY** | | | | | | | | | | |
| Heck et al, USA [35] | 1992 | Retrospective | 9 patients (12 TKAs) were time-matched to patients who underwent TKA within 3 months of the date of the arthroplasty. All operations carried out by a single surgeon | Mean (range): Cases: 67.4 (60–85) Controls: 73.5 (48–84) | Not given | OA, RA, post-traumatic arthritis and systemic lupus erythematosus | Not given | 6 years (0.75–9.6) | TKA revision surgery due to gross polyethylene failure defined as "polyethylene fracture or complete wear-through resulting in unintended prosthetic articulation with metal or bone" | - / High |
| Jones et al, USA [36] | 2004 | CC | 64 cases (primary TKA and revision) and 125 controls (primary TKA only) that met the eligibility criteria, of which 38 cases and 52 controls enrolled. Finally, 26 cases with TKA performed between October 1999 and September 2000 and 26 controls were individually matched for sex, age (± 5 years), unilateral or bilateral procedure and date of TKA (± 3 years). Operations performed by 12 orthopaedic surgeons across 4 hospitals | Mean (± SD): 70.5 (± 8.9). Range (47–85) | Cases and controls: Women: 17 (65%) and Men: 9 (35%) | Primary TKA: Bi or tri-compartmental knee OA | Cemented components, cases vs controls: femoral component (23% vs 69%), tibial component (58% vs 100%), patellar component (73% vs 100%) | Mean (SD):5 years (± 2.3), range (2–11) | Revision of either the tibial or femoral component occurring at a minimum of 2 years post-TKA due to aseptic loosening or mechanical failure | ++ / Low |
| Ponzio et al, USA [37] | 2018 | Retrospective | 5,328 patients from an institutional knee arthroplasty registry who underwent unilateral primary TKA between May 2007-February 2012. In total 1,008 active people and 1,008 inactive people were matched for age (±10 years), sex, BMI (5 ± kg/m²), ASA physical status and Charnley score | Mean (± SD): 66.3 ± 9.0 in the inactive group Mean (± SD): 66.3 ± 9.1 in active group | Men: 1,140 (56.6%) Women: 876 (43.5%) | Primary OA | Not given | 5 to 10 years post-operation Mean time to revision: 2.5 years, range (1.7 months– 8.2 years) in the active group, and 2.7 years (range 10.3 months– 6.8 years) for inactive group | All revision procedures identified from the database using Current Procedural Terminology (CPT) codes, regardless of the indication Indication for revision confirmed by chart review of the operative reports | + / Moderate |

*(Continued)*

**Table 1.** (Continued)

| Author, Country | Year | Study design | Number of participants | Age (years) at -primary arthroplasty | Gender | Indication for arthroplasty | Fixation technique | Duration of follow-up / mean time to revision | Definition of revision | Quality[a] / Risk of bias |
|---|---|---|---|---|---|---|---|---|---|---|
| Crawford et al, USA [38] | 2020 | Retrospective | 1,611 people (2,038 primary TKAs) with a minimum follow-up of 5 years post-operation and revision TKA procedures performed within the first 5 years post-TKA Participants identified from the author's institutional arthroplasty registry operated between 2003 and 2007 by two surgeons | Mean: 64.9 in the "low activity (LA)" group Mean: 62.3 in the "high activity (HA)" group | Men: 330 (27%) in the LA and 383 (46%) in the HA group Women: 880 (73%) in the LA, and 445 (54%) in the HA group | Not specified | Cemented | Mean: 11.4 years, range (5.1–15.9) / SD (±1.9) Mean (range) time to revision (years) for aseptic loosening or instability: 6.7 (0.9–12.7) in LA and 5.8 in HA group | TKA failure defined as revision of any component of the prosthesis | + / Moderate |

BMI: body mass index; CC: case control study; FU: follow-up; OA: osteoarthritis; RA: rheumatoid arthritis; THA: total hip arthroplasty; TKA: total knee arthroplasty; UCLA: University of California Los Angeles activity scale.

[a] Quality assessed as: high ++, acceptable +, poor 0, very poor—.

revision THA performed for any reason" [26]. For others, it was specifically described as a failure of the femoral acetabular component [29]. Five studies only included hip revision procedures for aseptic loosening [27, 30–33]. There was disparity amongst these studies, however: one focused on aseptic loosening of the femoral component [30] and another on aseptic loosening of either the femoral, acetabular, or both components [32].

## Occupation and occupational activities

Findings from studies that examined the risk of arthroplasty revision surgery and occupation or occupational activities are presented in Table 2.

**Pre-operative exposure to occupational activities.**   Three cohort studies assessed preoperative occupational exposures and the risk of revision hip arthroplasty [29–31]. The first study, by Maurer *et al*, rated of moderate quality, [30] categorised male recipients according to the nature of the physical activity performed in their pre-operative job: "no (or little) physical stress" as compared with"physically stressful or agricultural work". Unfortunately, the criteria by which the categories were defined were not stated. The rates of revision surgery were then compared amongst male THA recipients exposed to physical stress/ in farm work, those exposed to no (or little physical stress), and all female THA recipients. The authors reported that, compared with women, men had a 3-fold increased risk of THA revision when exposed to little or no physical stress (RR: 3.15 95%CI 1.70–5.80), and a 5-fold increased risk when exposed to physical stress/agricultural work (RR: 5.24 95%CI 2.80–9.80). The duration of follow-up varied between 5 and 10 years according to the type of stem implant received.

The second study (rated poor quality), by Inoue and colleagues [29] reported a higher risk of hip revision at a mean follow-up of 7.5 years post-THA for those working in agriculture at the time of THA compared with those not working in agriculture. The risk was highest for women working in agriculture compared with women not working in agriculture (RR:3.09, p = 0.04).

In the third study (rated acceptable quality), Flugsrud *et al*. [31] used occupational exposure collected during a cardiovascular screening assessment, carried out a median of 15 years before THA, and at least 6 years pre-operatively (in 95% of the cases). The mean age at primary THA was 63 years, whereas the mean age at censoring or event (revision) was 68 years. The authors found no association between either intensive, intermediate or moderate physical activity at work and the risk of revision for aseptic loosening of the cup or stem when compared with sedentary work.

A case-control study [28] (rated poor quality) found that, among women, exposure to self-reported "heavy work" before and after arthroplasty was associated with higher rates of THA revision (OR: 1.9, 95%CI 1.2–3.2). In terms of occupation type, they reported that women in health service jobs and those performing domestic work were at higher risk of revision surgery compared with women doing domestic work only (OR: 2.5, 95%CI 1.2–5.1). Other job titles (i.e., women in industry, engineering or construction work) were not found at higher risk.

Cumulatively, three of these studies provide some low-quality evidence that individuals doing physically-demanding work at least at the time of their primary THA (and presumably in many cases also after the surgery) may have a greater risk of subsequent revision. In particular, "heavy" work, agriculture and, in women, health services work, appeared to increase the risk. However, none of these studies provided complete information as to how many THA recipients (male or female) actually returned to their pre-operative occupation after surgery, and whether or not it was at the same or lower intensity compared with pre-operatively. In addition, one acceptable quality study found no association [31].

**Post-operative exposure to occupational activities.**   Only one study clearly measured only post-operative occupational exposures and it was in relation to the risk of revision after

Table 2. Findings from the studies assessing occupational activities and risk of lower limb revision arthroplasty by year of publication.

| Author | Number of participants | Exposure measurement timing | Occupation availability pre and/or post operation | Occupation assessment | Adjusted for | Risk estimate 95% CI |
|---|---|---|---|---|---|---|
| Espehaug et al [28] | 536 (primary operations and reoperations) and 1.092 controls (primary operation) | Not specified | Poorly described. Heavy physical work included in the analysis reported as: "previous exposure or, when relevant, exposure at follow-up" | A mail survey captured self-reported occupation, employed or not (yes/no), and whether the job involved "doing heavy physical work" (yes/no) | Covariates used to match cases and controls (age, date of THA and bilaterality) Extra analyses performed to avoid confounding using type of cement, prosthesis and use of antibiotic prophylaxis | Heavy work pre/post-THA yes vs no: OR: 1.5 (95% CI 1.1–2.2) overall OR; 1.1 (95% CI 0.7–2.0) in men OR; 1.9 (95% CI 1.2–3.2) in women Occupation ± domestic work vs domestic work (ref) among women: Industry/engineering/ construction and domestic work vs ref: OR; 2.0 (95% CI 0.7–5.7) Health service work vs ref: OR; 2.1 (95% CI 1.0–4.8) Health-service work and domestic work vs ref: OR; 2.5 (95% CI 1.2–5.1) Agriculture/ forestry /at sea and domestic work vs ref: OR; 1.7 (95% CI 0.9–3.3) Office/trade/hotel/service and domestic work vs ref: OR;1.4 (95% CI 0.8–2.2) Other combinations and domestic work vs ref: OR; 1.5 (95% CI 0.9–2.3) |
| Inoue et al [29] | 28 radiographic failures, of which 19 had undergone THA revision procedure | Exposure taken on admission from medical records | Pre-primary THA (at the time of the operation) | Occupation obtained at the time of the operation in a "descriptive manner" Includes whether worked in agriculture or not- but no specific details on occupation No details on the nurse interview questions or tool used to collect data | Age, sex, diagnosis, cementing technique | Working in agriculture, yes vs no: Overall RR; 2.85 (95% CI 1.10–7.36) p = 0.03 Men: RR; 2.37, p = 0.40 Women: RR; 3.09 p = 0.04 |
| Maurer et al [30] | 589 consecutives primary THAs with 88 revisions due to aseptic loosening of the stem: 4 CoCrNi alloy stem, 32 Titanium SS 77 stem and 52 Titanium SLS stem | Exposure measured at the time of the operation | Physical stress at work recorded at the time of the implantation. | Farming work considered as physical stress | Age, stem type, stem size | Men with little physical stress at work vs women: RR; 3.15 (95% CI 1.70–5.80) Men with physical stress or in farming work vs women: RR; 5.24 (95% CI 2.80–9.80) |

(*Continued*)

**Table 2.** (Continued)

| Author | Number of participants | Exposure measurement timing | Occupation availability pre and/or post operation | Occupation assessment | Adjusted for | Risk estimate 95% CI |
|---|---|---|---|---|---|---|
| Jones et al [36] | 26 cases (primary TKA and revision) and 26 controls (primary TKA only) | Not specified (TKAs performed between 1999 and 2000) | Post- primary TKA (second year after primary operation onwards) | Information collected by a structured phone interview using the Modifiable Activity Questionnaire (MAQ) Occupation with a metabolic equivalent (MET) $\geq 7$ considered as high intensity. | Covariates used to match cases and controls | Physical activity at work: OR; 0.99 (95% CI 0.99–1.01) High intensity physical activity at work: OR; 1.0 (95% CI 0.99–1.01) |
| Flugsrud et al [31] | 165 THA revision procedures due to aseptic loosening: 59 for stems, 49 for cups and 57 for both | Cardiovascular screening carried out from 1977–1983 | Pre- primary THA | Physical activity at work collected in a cardiovascular screening carried out during 1977–1983 (pre-THA) using the Saltin-Grimby scale Participants' job classified as: i) sedentary (mostly sedentary work), ii) moderate (work related to much walking), iii) intermediate (work involving much walking and lifting), or iv) intensive (heavy manual work) [39] | Age at screening, height, BMI, physical activity at work, leisure activities, marital status, smoking and implant category. | Physical activity at work RR (95% CI): *Intensive vs sedentary:* Men: 0.6 (0.2–1.6) for the cup and 0.6 (0.3–1.5) for the stem Women: 0.9 (0.3–3.0) for the cup and 0.6 (0.1–2.5) for the stem *Intermediate vs sedentary:* Men: 0.6 (0.2–2.0) for the cup and 0.7 (0.3–1.9) for the stem Women: 1.0 (0.4–2.4) for the cup and; 0.9 (0.3–2.7) for the stem *Moderate vs sedentary:* Men: 0.6 (0.2–1.8) for the cup and 0.8 (0.3–2.0) for the stem Women: 0.7 (0.3–1.5) for the cup and 1.3 (0.5–3.0) for the stem |

BMI: body mass index; THA: total hip arthroplasty; TKA: total knee arthroplasty.

primary TKA. Jones *et al.*, in a study rated high quality, investigated the risk of revision TKA in relation to historical occupational activity over an average period of 4 years (SD ± 2) [36]. They found no association between working in occupations with a higher number of metabolic equivalent (MET) hours of physical activity/week and the risk of primary TKA revision.

## Leisure-time physical activities (LTPA)

Five studies examined the effect of leisure-time activities on the risk of revision of hip and knee arthroplasty. Table 3 summarises the results for exposure to LTPA and risk of lower limb arthroplasty revision.

**Pre-operative leisure-time physical activity.** Three studies evaluated pre-operative sports or LTPA in relation to risk of revision after hip arthroplasty [28, 29, 31]. Flugsrud *et al.* [31] found that men who participated in intermediate/intensive physical activity before THA were at increased risk of cup revision compared with sedentary men (RR: 4.8, 95%CI 1.3–18.2) [31]. In contrast, Inoue *et al*, rating pre-operative exposure to LTPA as "none" or "some" activity found no association between these levels of recreational activities before THA and risk of subsequent arthroplasty failure (RR:0.89, 95%CI 0.40–1.98) [29]. Unfortunately, neither of these papers provided specific information about the likelihood that THA recipients returned to the same level of physical activity post-operatively, hindering interpretation of these results.

Espehaug *et al.* [28], rated as acceptable quality, collected data about recreational activities performed before the hip symptoms started and found no association between participation in competitive sports before the primary operation and risk of THA revision (OR: 1.3, 95%CI 0.9–2.1). However, in terms of frequency of recreational activity, men (but not women) doing exercise on a regular basis (weekly) before THA were found to be at increased risk of a THA revision (OR: 2.6, 95%CI 1.4–4.7) compared with those not exercising on a regular basis.

**Post-operative exposure to leisure-time physical activity.** Espehaug *et al* found that amongst men and women reporting regular exercise post-THA, there was no associated increased risk of revision post-THA (OR:0.8 95%CI 0.5–1.0) [28]. Dubs *et al.* [26], in a study rated of poor quality, collected data on sporting activities both pre- and post-THA, but the precise definition of practising sport "regularly" was not stated, nor were the type of questions or scale used to collect the data. They found no significant effect of participation in sport on the risk of THA but did find a strong tendency for the active group to be less likely to need hip revision during the follow-up (14.3% non-active vs 1.6% sports participators).

Only one paper assessed post-operative LTPA in relation to the risk of TKA revision. Jones and colleagues [36] recorded the average number of hours that people engaged in 39 leisure and sport activities post-operatively after TKA. Their results showed no increased risk of TKA revision in participants doing high-intensity leisure activities (OR: 0.96, 95%CI 0.88–1.05).

Taken together, there is no convincing evidence that post-operative LTPA increases the risk of revision after THA or TKA.

## Total physical activity

Of the thirteen studies retrieved, eight evaluated the effect of level of total physical activity or a combination of work and leisure activities, on the risk of revision surgery: four after knee arthroplasty and four after hip arthroplasty (Table 4).

**Pre-operative total physical activity.** Two studies investigated the risk of subsequent revision TKA based upon the total activity exposure reported by participants at the time of their primary surgery. In one study [37], rated of moderate quality, the Lower-Extremity Activity Scale (LEAS) was used to classify participants as physically "active" (LEAS 13 to 18) or "inactive" (LEAS 7 to 12) at the time of their TKA. These investigators found that at 2 years,

**Table 3. Findings from the studies examining exposure to leisure-time physical activity and the risk of lower limb revision arthroplasty by year of publication.**

| Author | Number of events | Exposure measurement timing | LTPA availability pre and/or post operation | Physical activity assessment | Adjusted for | Risk estimate 95% CI |
|---|---|---|---|---|---|---|
| Dubs *et al.* [26] | 9 (5.9%) THA implants failed (8 patients) | Not specified | Pre and post-THA | Sports activity (regular/none) recorded retrospectively using a self-administered questionnaire | Not applicable | 7 (14.3%) participants who did not practise sport post-THA needed revision, 1 (1.6%) participant who practised sport regularly required revision. Estimated risk for participants doing sports pre and post-THA vs participants less active/ not doing sports calculated from figures from the study: RR; 0.13 (95% CI 0.02–1.02) |
| Espehaug *et al.* [28] | 536 (primary operations and reoperation) and 1,092 controls (primary operation) | Not specified | Before the first hip symptoms and post-THA | Physical activity (sports and recreation) was measured as participation in competitive sports (yes/no) and weekly exercise (yes/no) "before the first hip symptoms" and post-THA | Covariates used to match cases and controls (date of THA and bilaterality). Extra analyses performed to avoid confounding using type of cement, prosthesis and use of antibiotic prophylaxis | Regular vs no regular exercise: *Before THA* Overall: OR 1.6 (95%CI 1.1–2.2) Men: OR: 2.6 (95% CI 1.4–4.7) Women: OR: 1.2 (95% CI 0.8–1.8) *Post-THA* Overall OR: 0.8 (95% CI 0.5–1.0) Men OR: 0.7 (95% CI 0.4–1.2) Women OR: 0.8 (95% CI 0.5–1.2) Active competitive sport before-THA vs no: Overall OR: 1.3 (95%CI 0.9–2.1) Men OR: 1.1 (95% CI 0.6–1.9) Women OR: 1.8 (95% CI 0.9–3.5) |
| Inoue *et al.* [29] | 28 radiographic failures, of which 19 underwent THA revision procedures | Exposure taken on admission from medical records | Pre-primary THA (at the time of operation) | Recreational activities recorded at the time of the operation in a "descriptive manner". No details on the nurse interview questions or tool used to collect data | Age, sex, diagnosis, cementing technique | Recreational activity: Some activity vs none: RR 0.89 (95% CI 0.40–1.98) p = 0.77 |
| Jones *et al.* [36] | 26 cases (primary TKA and revision) and 26 controls (primary TKA only) | Not specified | Post-primary TKA (from second year post-arthroplasty onwards) | Information collected from the second year post-TKA by a structured phone interview using the Modifiable Activity Questionnaire (MAQ). Leisure activities with a metabolic equivalent (MET) ≥6 considered as high intensity | Covariates used to match cases and controls | Leisure activities: OR; 0.99 (95% CI 0.99–1.02) High intensity leisure activities: OR; 0.96 (95% CI 0.88–1.05) |
| Flugsrud *et al.* [31] | 165 THA revision procedures due to aseptic loosening: 59 for stems, 49 for cups and 57 for both. | Not specified | Pre-primary THA | Leisure activities recorded in a cardiovascular screening carried out during 1977–1983 (pre-THA) using the Saltin-Grimby scale. Participant's leisure grouped as: sedentary (sedentary activities), moderate (walking or moving around at least 4 hours/week), intermediate (recreational athletics 4 hours/week) and intensive (hard training or athletic competitions, regularly and several times a week) [39] | Age at screening, height, BMI, physical activity at work, leisure activities, marital status, smoking and implant category. | Intensive & intermediate leisure activity vs sedentary RR (95% CI): Men: 4.8 (1.3–18.2) for the cup and 1.1 (0.5–2.8) for the stem Women: 1.6 (0.6–4.1) for the cup and 1.3 (0.5–3.4) for the stem Moderate leisure activity vs sedentary RR (95% CI): Men: 3.1 (0.8–11.8) for the cup and 0.9 (0.4–2.2) for the stem Women: 0.7 (0.4–1.5) for the cup and 0.6 (0.3–1.2) for the stem |

BMI: body mass index; LTPA: leisure-time physical activity; THA: total hip arthroplasty; TKA: total knee arthroplasty.

**Table 4. Findings from studies evaluating total exposure to physical activity (not separating occupational and leisure-time exposure) and the risk of lower limb revision surgery by year of publication.**

| Author | Number of events | Exposure measurement timing | Physical activity availability pre and/or post operation | Physical activity (PA) assessment | Adjusted for | Risk estimate 95% CI and p-value |
|---|---|---|---|---|---|---|
| Kilgus et al. [27] | 42 (6%) THAs revised in the less active group and 7 (28%) in the more active group. | Not specified | Post- THA | Physical activity assessed using medical notes, examining or contacting patients to evaluate their participation in either heavy work or sports post-THA. Participants were classified according to aetiology (OA versus non-OA) into: a) active if they participated regularly in heavy labour for several years and/or sports post-THA or b) less active if they did not participate regularly in heavy labour or sports | Age, length of FU period, diagnosis and surgical technique | The overall revision rates were as follows: active group, 28%; and less active group, 6%. Patients engaged in sports post-THA had over twice the risk of revision for aseptic loosening compared with less active patients |
| Heck et al. [35] | 12 TKA revisions in 9 participants (cases) and a time-matched control group | Not specified | At the time of TKA | Level of physical activity grouped using a modification of the Old-age, Survivors, and Disability Insurance (OASDI) activity level scoring system. ranging from 0 (in nursing home with full time care) to 7 (very heavy labour) Participants were classified as: sedentary (level 0 to 3) or as performing at a higher activity level (level 4 to 7) | Not applicable | Physical activity level in revised patients was higher compared with that reported by patients not requiring revision, p = 0.023 |
| Jones et al. [36] | 26 cases (primary TKA and revision) and 26 controls (primary TKA only) | Not specified | Post-primary TKA (from second year post-arthroplasty onwards) | A combination of the historical leisure and occupational activity using the Modifiable Activity Questionnaire (MAQ) | Covariates used to match cases and controls | <u>Leisure activities and work:</u> OR:0.99 (95% CI 0.99–1.01) Total historical physical activity (high vs low level): OR: 0.67 (95% CI 0.67–1.93) |
| Lübbeke et al. [32] | Femoral osteolysis developed in: 5.4% (9/166) of the low activity patients, 7.5% (21/279) of the moderate activity patients and 24.1% (14/58) of the high activity patients. Of the 44 patients with femoral osteolysis, 4 were revised: 2 in high activity group, 2 in moderate activity group and none in low activity group | Physical activity assessed in two cohorts of patients: at 5 years review post-THA (2001 to 2003), and at 10 years review post-THA (1996 to 1998) | Post- primary THA | Level of physical activity assessed by the UCLA activity scale post-THA and grouped as: UCLA 1–4 (low activity) UCLA 5–7 (moderate activity) UCLA 8–10 (high activity) More accurate information on participation in recreational and/or sport activities obtained by questioning patients | Not applicable | The risk of revision for the femoral component increased significantly with increasing levels of physical activity post-THA (p = 0.023). |
| Ollivier et al. [33] | 7 patients revised for aseptic loosening; 6 in the high impact activities group (2 for the acetabular component and 4 for the femoral component) and 1 in the low activity group due to loosening of the acetabular component | Questionnaire at a minimum of 10 years post-operation | Post- primary THA | Level of physical activity assessed by self-administered questionnaire and the UCLA activity scale. Participants grouped as: High impact UCLA 9–10 Low impact: UCLA 1 to 4 | Not specified | <u>High impact sport vs low impact activities:</u> OR: 3.64 (95% CI, 1.49–8.9) |

*(Continued)*

**Table 4.** (Continued)

| Author | Number of events | Exposure measurement timing | Physical activity availability pre and/or post operation | Physical activity (PA) assessment | Adjusted for | Risk estimate 95% CI and p-value |
|---|---|---|---|---|---|---|
| Delfin et al. [34] | 27 cases (THA and subsequent revision) matched with 27 controls (THA without revision surgery) | Questionnaire sent out in November 2014 | Post-primary THA | Physical activity assessed by a modified UCLA activity scale to recall activity level after THA. Frequency of physical activity measured on a scale ranging from 0 "practically no physical activity at all" to 5 "vigorous physically active at least twice a week" | Covariates used to match cases and controls | 81.5% of the revisions were due to aseptic loosening and 18.5% due to dislocation. UCLA score ≥ 5 in 56% of the cases and 67% of the controls. Risk for revision: UCLA score: OR; 0.96 (95% CI 0.73–1.3) Frequency of physical activity: OR; 0.46 (95% CI 0.12–1.84) |
| Ponzio et al. [37] | 32 participants out of 1,008 in the active group, and 16 participants out of 1,008 in the inactive group | Questionnaire for physical activity completed pre-TKA | Pre-primary TKA | Regular daily activity assessed by Lower-extremity Activity Scale (LEAS) Participants grouped as: Inactive: LEAS 7–12 Active: LEAS 13–18 | Not applicable | Revision rate at 5 to 10 years post-TKA: For aseptic loosening: 8 in the active group (25%) and 1 (6.3%) in the inactive group (p = 0.238), For osteolysis and wear: 3 (9.4%) in the active group and 0 (0%) in the inactive group (p = 0.541) For instability: 9 (28.1%) in the active group and 5 (51.3%) in the inactive group (p = 0.999) For stiffness: 4 (12.5%) in the active group and 4 (25%) in the inactive group (p = 0.413) For fracture: 2 (6.3%) in the active group and 0 in the inactive group (p = 0.546) For malalignment: 1 (3.1%) in the active group and 0 in the inactive group (p = 0.999) For patellar loosening: 1 (3.1%) in the active group and 1 (12.5%) in the inactive group (p = 0.254) For polyethylene dissociation: 1 (3.1%) in the active group and 0 in the inactive group (p = 0.999) For all cause: 32 (3.2%) in the active group and 16 (1.2%) in the inactive group (p = 0.019) |
| Crawford et al. [38] | 49 out of 1,210 in the low activity group 14 out of 828 in the high activity group | Not detailed when exposure was measured at follow-up | Post-primary TKA | Physical activity based on the UCLA activity score Participants grouped as: Low activity: UCLA 1–5 High activity: UCLA 6–10 | Not applicable | Kaplan-Meier aseptic survival rate at 12 years FU: 98.4% (95%CI 97.9–98.9) for the "high activity" group and 96.3% (95%CI 95.6–97) for the "low activity" group (p = 0.02) |

BMI: body mass index; CC: case control study; FU: follow-up; OA: osteoarthritis; THA: total hip arthroplasty; TKA: total knee arthroplasty; UCLA: University of California Los Angeles activity scale.

69.5% of the inactive patients and 27.3% of the active patients improved their baseline activity level (p<0.0001). In the crude analyses, aseptic failure rate was 6.3% in the low activity group and 25% in the high-activity group (p = 0.238). At 5 to 10 years' post-operation, the revision rate for all causes (including infection) was different between the active and inactive groups (p = 0.019), whereas revision rate for all non-infective causes was not statistically significantly different between active and inactive groups. In the second study (rated of very poor quality) activity levels were categorised according to the Old-age, Survivors, and Disability Insurance (OASDI) classification, grouping OASDI activity levels 0 to 3 as "sedentary" and 4–7 as "higher activity levels" [35]. The authors reported that participants undergoing TKA revision had a higher activity level at the time of the primary operation than those who did not require revision. Unfortunately, the paper provided no indication as to how active participants were after their primary operation.

**Post-operative total physical activity.** Three studies used the University of California Los Angeles (UCLA) activity scale to capture the level of total physical activity post-THA in relation to risk of revision. The first study, of acceptable quality, [32] showed that, amongst 44 of 433 patients who developed femoral osteolysis, revision for aseptic loosening was more likely with increasing levels of UCLA activity post-operatively measured at 5- or 10-year follow-up. The second study [33], also of acceptable quality, reported that people doing UCLA-rated high impact activities (this includes high-impact sports such as jogging and/or heavy labour [40]) were three times more likely to undergo hip revision compared with those who engaged only in low impact activities (OR:3.64, 95%CI 1.49–8.9) [33]. The third study, with a high methodological quality score, found no association between either the level or frequency of any physical activity post-THA, and the risk of revision [34].

Another study, rated poor quality, [27] classified participants as "active" or "less active", defining them as "active" if they either regularly participated in sports or heavy labour for a period of several years following their total THA. The authors reported that the activity information was obtained from medical records, or by phone and letter contact, and/or patient examination (Table 4). The authors reported a more than doubling of the risk of revision amongst those who were active when compared with the less active group.

In their study of risk of revision after TKA, Jones *et al* reported no difference between the level or frequency of historical physical activities (both work and leisure) among people who had undergone TKA revision compared with those who had not, and no association between high levels of historical physical activity and the risk of revision surgery [36]. However, another study rated [38] of moderate quality found that survival rate due to aseptic loosening was better amongst knee arthroplasty recipients with a high level of physical activity (UCLA 6 to 10) compared with those with a low level of physical activity (98.4% (95%CI 97.9–98.9) vs 96.3% (95%CI 95.6–97), p = 0.02). This study included participants followed-up for a minimum of 5 years but also those who underwent a revision procedure within 5 years.

In summary therefore, we found conflicting evidence with respect to total post-operative activity levels and an increased risk of revision after hip and knee arthroplasty.

## Discussion

This systematic review examined the evidence about exposure to high-intensity or physically-demanding activities either at work or in leisure-time and the risk of hip or knee revision surgery. From 11,307 studies identified as of interest, 13 fulfilled our inclusion/exclusion criteria. Amongst five studies exploring the role of occupation (or occupational activities) after hip arthroplasty, two reported a positive association with pre-operative farming [29, 30]; one reported an increased risk with heavy physical work (pre and post-THA) [28], and one no

effect [31]. The only study of occupational activities after TKA found no association [36]. For revision THA with sports and LTPA participation, 4 studies were inconsistent: two found increased risk [28, 31], (only amongst men) [28]; one found no effect [29]; and one (poor-quality) study found reduced risk [26]. Three studies evaluated THA revision and total activities (work and leisure) using the same measurement tool (UCLA) but were also conflicting: one suggested an increased risk with increasing activity [32]; one suggested an increased risk with high-impact sport [33]; and the best quality study found no effect for either level or intensity of activities [34]. Another (weak) study measured total activities with a different tool and reported a doubling of risk of revision THA [27]. For LTPA after TKA, one study found no association [36], another found better implant survival with more physical activity [38], one found that total physical activity at the time of TKA increased risk of revision for all causes, but not after exclusion of those performed for infectious causes [37] and another (poor quality) study reported that LTPA increased the risk of revision [35]. Taken together, we found a heterogeneous literature unsuitable for pooling for quantitative synthesis. The evidence is unconvincing for an increased risk of revision after hip or knee arthroplasty associated with LTPA, and although there is some evidence for increased risk of revision THA with physically-demanding work, more research is required using standardised methodology. In particular, more studies are needed after TKA.

We experienced methodological challenges in assessing the evidence. Firstly, to address our research question, the exposure to physical activity should ideally be measured both before and after lower limb arthroplasty, since accounting only for pre-operative work or leisure physical activity may lead to misclassification of the post-operative exposure. People who were very active pre-operatively may not necessarily be able to achieve the same level of activity post-operatively and vice versa. Certainly, pooled data from 4 studies which examined engagement with sport amongst THA patients indicated that, overall, 18% of people did not resume such activities post-operatively [41]. Additionally, three reviews found a reduction in the intensity or impact of sports participation among patients post-operatively [41–43]. Generally, most people working pre-operatively return to work after arthroplasty [41], but they may move to a different, occupation [44, 45], involving less-physically demanding activities. In order to address our research question more effectively, researchers need to more clearly collect actual exposure data using reliable methods. Secondly, the definition of revision arthroplasty, and the indication for carrying out revision, varied between the studies. In the majority, the main indication for revision was aseptic loosening, but a few studies used a broader definition [26, 28, 34]. Ollivier *et al*, for example, defined implant failure as "hip revision in the presence of radiographic signs of aseptic loosening" [33]. Thirdly, investigators in these studies used a wide variety of methods and measurement tools to collect information about exposure to work and leisure activities, which precluded comparison of results between studies, not least because each instrument (often non-standardised) referred to a different recall period of activity. Those studies which attempted to measure occupational exposures used job title or subjective assessments such as "heavy work". Neither of these types of methods has good reliability or validity for the assessment of true occupational exposure [46] and more high-quality data collection in this area is desperately needed. Equally, participation in LTPA can vary markedly, even when individuals ostensibly report the same sporting activity e.g., tennis or running/jogging. Future studies need to include more detailed measures which either better identify specific loading of the joints in question or at least more accurately quantify the intensity and duration of the exposures over time. Use of the UCLA activity score might have facilitated comparison of the results from three of the retrieved studies but, as already observed, researchers reported their results differently. Arguably an "overall" activity level is less helpful for advising patients compared to a separate assessment of work and LTPA.

Unfortunately, there are no clinical or consensus guidelines about the resumption of post-operative occupational activities (likely because of the lack of evidence). A recent qualitative study found that surgeons tend to assume that most arthroplasty recipients have retired and only a small minority are employed who wish or need to RTW post-operatively [47]. However, they acknowledged that they are likely to see an increasing number of patients who expect to return to work post-operatively and they agreed that they currently provide limited occupational advice to patients, which was largely based upon whether the individual undertook desk-based work as compared with any other type of work [47]. In some cases, they suggested that they might advise patients with manual jobs to consider changing their occupation, particularly if specific activities were involved e.g., kneeling. It is widely acknowledged that work is important to health and financial stability [48]. Many governments have made legislative changes to encourage people to work to older ages so that in the UK, for example, people will only be entitled to claim their state pension at 67 years of age or above rising to 68 for those born after April 1978. Therefore, working post-arthroplasty is set to become a more common phenomenon and the current review reveals the size of the evidence gap and the growing need for carefully-designed research that accurately measures post-operative occupational activities and the risk of revision in order to enable surgeons to give constructive advice to future patients.

Interpretation of the findings of this review must consider some limitations. We limited our search to include only those studies published following a peer-reviewed process, choosing to exclude articles published in the grey literature. Whilst this may increase the risk that our findings are affected by publication bias, the likelihood of this is somewhat reduced in that relatively few of the included papers investigated the role of occupational and/or leisure-time activities as their primary factors of interest. Indeed, if anything, leisure-time activities and sports participation were more often included in the title of papers and our findings show that, despite this, we could not find convincing evidence of their association with the risk of revision surgery. The authors acknowledge that omission of the grey literature here may have limited the comprehensiveness of our review. Additionally, the search was limited to publications in English or Spanish and therefore may have missed studies published in other languages, although key papers are more likely to be published in English. Unfortunately, we were only able to perform a narrative synthesis of the evidence rather than a quantitative analysis for a range of reasons including: the heterogeneity of the time frame of measurement of occupational/physical activities (pre-operative, perioperative and post-operative) the wide variation in the methods of assessment of these activities; the small number of studies that addressed leisure-time and occupational activities separately; and the variability of the duration of post-operative follow-up (ranged from a minimum of 4.9 years to a maximum of 11 years). Furthermore, unfortunately, revision was only a secondary outcome in three of the included studies [30, 32, 37]. As the risk of revision has declined with improved materials and surgical techniques, so statistical power to detect risk is diminished unless large-scale studies are carried out, with a very long duration of follow-up. It is for this reason that arthroplasty registers have been set up [49] and these could provide an ideal framework for investigating the current research questions. Despite these limitations, the current review is, to the best of our knowledge, the first to examine the effect of occupation and leisure activities on the risk of lower limb arthroplasty revision.

## Conclusion

In summary, the findings from this review highlighted the paucity of relevant studies on this research question, especially for revision surgery after TKA. Many studies only assessed relevant exposure pre-operatively, which is likely to be of limited relevance to post-operative

activities. Based on the limited evidence identified, occupation and leisure-time physical activity do not convincingly increase the risk of revision after hip or knee arthroplasty. Given the lack of evidence and the inconsistencies found, more research is needed to assess the risk of mechanically loading the replaced hip or knee following joint arthroplasty, and, in particular, to investigate the impact of return to physically-demanding occupational activities, given that increasing numbers of people will want and need to return to work post-arthroplasty.

## Supporting information

**S1 File. MeSH terms and keywords used in the search.**
(DOCX)

**S1 Table. Quality assessment of cohort studies.**
(DOCX)

**S2 Table. Quality assessment of case-control studies.**
(DOCX)

## Author Contributions

**Conceptualization:** E. Clare Harris, Karen Walker-Bone.

**Formal analysis:** Elena Zaballa, E. Clare Harris, Catherine H. Linaker, Karen Walker-Bone.

**Funding acquisition:** Cyrus Cooper, Karen Walker-Bone.

**Methodology:** Elena Zaballa, E. Clare Harris.

**Supervision:** E. Clare Harris, Karen Walker-Bone.

**Writing – original draft:** Elena Zaballa.

**Writing – review & editing:** Elena Zaballa, E. Clare Harris, Cyrus Cooper, Catherine H. Linaker, Karen Walker-Bone.

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
