## [Decision Letter · Decision Letter 0]

1 Dec 2021

PONE-D-21-27689Risk of revision arthroplasty surgery after exposure to physically demanding occupational or leisure activities: a systematic reviewPLOS ONE

Dear Dr. Zaballa Lasala,

Thank you for submitting your manuscript to PLOS ONE. After careful consideration, we feel that it has merit but does not fully meet PLOS ONE’s publication criteria as it currently stands. Therefore, we invite you to submit a revised version of the manuscript that addresses the points raised during the review process. There are some major concerns that need to be addressed in relation to the requirements for a systematic review. 

We look forward to receiving your revised manuscript.

Kind regards,

John Leicester Williams, Ph.D.

Academic Editor

PLOS ONE

Journal Requirements:

2. Please provide a table reporting in detail the results of your quality assessment, showing how each included study scored on every item of the modified Scottish Intercollegiate Guidelines Network (SIGN) checklist and Assessment of Quality in Lower Limb Arthroplasty checklist.

Reviewers' comments:

Reviewer's Responses to Questions

**Comments to the Author**

1. Is the manuscript technically sound, and do the data support the conclusions?

Reviewer #1: Yes

Reviewer #2: Yes

Reviewer #3: Partly

2. Has the statistical analysis been performed appropriately and rigorously? 

Reviewer #1: N/A

Reviewer #2: N/A

Reviewer #3: N/A

3. Have the authors made all data underlying the findings in their manuscript fully available?

Reviewer #1: Yes

Reviewer #2: No

Reviewer #3: Yes

4. Is the manuscript presented in an intelligible fashion and written in standard English?

Reviewer #1: Yes

Reviewer #2: Yes

Reviewer #3: Yes

5. Review Comments to the Author

Reviewer #1: In this manuscript, authors utilized explicit, systematic methods to conduct a comprehensive systematic review to evaluate if exposure to physically demanding occupational or leisure activities increased the risk of revision arthroplasty surgery. There are some major concerns and a few minor concerns that should be addressed.

Major concerns:

1. Are there any specific reasons for including publications from peer-review journals only? As we know, statistically significant findings are more likely to be published. If only include peer-review publications, selection bias might exist in this study and lead to positive findings.

2. In the search strategy section, the authors mentioned that letters and notes were excluded. However, letters or notes might include data or information for this study. Thus this strategy might lead to selection bias as well.

3. Line484, comments on the potential impact of each limitation should be discussed. For example, the impact of including peer-review publications on the result.

4. Quantitative analysis is critical for the assessment of the research questions. The authors should provide a reasonable explanation for why meta-analysis was not conducted.

Minor concerns:

1. In abstract, should have a method to assess the risk of bias

2. Line100 When using an acronym (PICO) for the first time, it must be spelled out in the text

3. In search strategy, the date when sources were the last search should be specified in the text.

4. Line 111 authors mentioned that the full paper would be checked if the conference abstract was found, but how did you address if no full paper was found?

5. Line192 When using an acronym (OA) for the first time, it must be spelled out in the text

6. According to “PRISMA 2020 explanation and elaboration: updated guidance and exemplars for reporting systematic reviews”, a general interpretation of the results should be provided. The first paragraph of the discussion is too detailed. A concise and informative paragraph might be more appropriate here.

Reviewer #2: General comments

Thank you for giving me the opportunity to review this manuscript. Well-performed systematic reviews have high clinical importance. My compliments to the authors for a well performed systematic review. My main concern is about how the discussion is lined out. Besides that, I only have a few comments.

Title

Appropriate

Abstract

I recommend not to use emphasising words like “very” in a scientific paper.

Introduction

Gives a good rationale for the study.

Materials and methods

Well described.

Results

Well written.

Discussion

In the first page on the discussion, all results are repeated, without any actual discussion. I recommend the authors to start with a short summary of main findings, and then discuss the results, in relation to other studies.

Page 31, line 443-454: this paragraph would be more relevant in the introduction.

Page 32, line 455-463: this part does not seem to have any relation with the results from the present study.

Page 32, line 465-480: introduction, not discussion

Conclusion

The first part in the conclusion is relevant. The last part, with suggestions for future research is not a conclusion and should be moved to discussion.

Tables and figures

Appropriate

References

Appropriate

Reviewer #3: Please find the attached file with my comments regarding the manuscript Risk of revision arthroplasty surgery after exposure to physically demanding occupational or leisure activities: a systematic review

6. PLOS authors have the option to publish the peer review history of their article (what does this mean?). If published, this will include your full peer review and any attached files.

Reviewer #1: No

Reviewer #2: No

Reviewer #3: No

---

## [Author Response · Author response to Decision Letter 0]

14 Jan 2022

Reviewer #1: In this manuscript, authors utilized explicit, systematic methods to conduct a comprehensive systematic review to evaluate if exposure to physically demanding occupational or leisure activities increased the risk of revision arthroplasty surgery. There are some major concerns and a few minor concerns that should be addressed.

Major concerns:

1. Are there any specific reasons for including publications from peer-review journals only? As we know, statistically significant findings are more likely to be published. If only include peer-review publications, selection bias might exist in this study and lead to positive findings. 

Thank you for your comment. We agree absolutely with the Reviewer’s important point about the risk of publication bias in general and its effect serving to over-emphasise effect sizes. However, as the Reviewer will have deduced, we found that most of the papers that met our eligibility criteria considered risk of revision surgery in relation to a range of possible risk factors and included occupational and/or leisure-time physical activities amongst them. If anything, more of the included papers featured leisure-time /sports activities in their title and, despite this, we could not find sufficient high-quality evidence of an adverse effect of these activities on the risk of revision surgery. We chose to concentrate on peer-reviewed journals considering that there might be an equally large, or even larger, risk of methodological weakness or bias amongst papers published therein. A brief search also confirmed that the grey literature includes very little information about occupational activities and risk of revision surgery.

In reference to the Reviewer’s point however, we have added this as a possible limitation In the Discussion section (Page 33 Line 473 of 'Manuscript' document):

“Interpretation of the findings of this review must consider some limitations. We limited our search to include only those studies published following a peer-reviewed process, choosing to exclude articles published in the grey literature. Whilst this may increase the risk that our findings are affected by publication bias, the likelihood of this is somewhat reduced in that relatively few of the included papers investigated the role of occupational and/or leisure-time activities as their primary factors of interest. Indeed, if anything, leisure-time activities and sports participation were more often included in the title of papers and our findings show that, despite this, we could not find convincing evidence of their association with the risk of revision surgery. The authors acknowledge that omission of the grey literature here may have limited the comprehensiveness of our review.”

2. In the search strategy section, the authors mentioned that letters and notes were excluded. However, letters or notes might include data or information for this study. Thus this strategy might lead to selection bias as well.

As above, this was a decision made priori and published in our protocol in PROSPERO. 

3. Line 484, comments on the potential impact of each limitation should be discussed. For example, the impact of including peer-review publications on the result.

Thank you. This has been included.

4. Quantitative analysis is critical for the assessment of the research questions. The authors should provide a reasonable explanation for why meta-analysis was not conducted.

We agree with the reviewer that pooling data from studies is the best way to address the research question. Unfortunately, a quantitative approach was not possible, for the reasons given in the Discussion section; mainly because exposure was measured at different time points in relation to the surgery and because the methods used to assess physical activity (work, non-work related and all activity) were very variable, revision was not always the primary outcome, and the length of follow-up varied widely. 

We have spelled these points out more clearly in the revised Limitations paragraph of the Discussion; Page 33 Line 473 of ‘Manuscript’ document

“Interpretation of the findings of this review must consider some limitations. We limited our search to include only those studies published following a peer-reviewed process, choosing to exclude articles published in the grey literature. Whilst this may increase the risk that our findings are affected by publication bias, the likelihood of this is somewhat reduced in that relatively few of the included papers investigated the role of occupational and/or leisure-time activities as their primary factors of interest. Indeed, if anything, leisure-time activities and sports participation were more often included in the title of papers and our findings show that, despite this, we could not find convincing evidence of their association with the risk of revision surgery. The authors acknowledge that omission of the grey literature here may have limited the comprehensiveness of our review. Additionally, the search was limited to publications in English or Spanish and therefore may have missed studies published in other languages, although key papers are more likely to be published in English. Unfortunately, we were only able to perform a narrative synthesis of the evidence rather than a quantitative analysis for a range of reasons including: the heterogeneity of the time frame of measurement of occupational/physical activities (pre-operative, perioperative and post-operative) the wide variation in the methods of assessment of these activities; the small number of studies that addressed leisure-time and occupational activities separately; and the variability of the duration of post-operative follow-up (ranged from a minimum of 4.9 years to a maximum of 11 years). Furthermore, unfortunately, revision was only a secondary outcome in three of the included studies [30, 32, 37]. As the risk of revision has declined with improved materials and surgical techniques, so statistical power to detect risk is diminished unless large-scale studies are carried out, with a very long duration of follow-up. It is for this reason that arthroplasty registers have been set up [49] and these could provide an ideal framework for investigating the current research questions. Despite these limitations, the current review is, to the best of our knowledge, the first to examine the effect of occupation and leisure activities on the risk of lower limb arthroplasty revision.”

Minor concerns:

1. In abstract, should have a method to assess the risk of bias. 

Thank you. We have included this as follows (Page 2 Line 38):

“We searched Medline, Embase and Scopus databases (1985 - July 2021) for original studies including primary lower limb arthroplasty recipients that gathered information on physically-demanding occupational and/or leisure activities and rates of revision arthroplasty. Methodological assessment was performed independently by two assessors using SIGN, AQUILA and STROBE. The protocol was registered in PROSPERO [CRD42017067728].”

2. Line100 When using an acronym (PICO) for the first time, it must be spelled out in the text

Thank you - this has been spelled out in full

Following the Population, Intervention, Comparison and Outcome (PICO) format

3. In search strategy, the date when sources were the last search should be specified in the text. 

Thank you for pointing this out. The date the last search was performed has been introduced in the text accordingly. Please see Page 6 Line 120 of ‘Manuscript’ document:

“Our search was conducted in three electronic databases: MEDLINE and Embase using the Ovid search engine, and in Scopus (S1 File), limited to studies published in peer-reviewed journals, from January 1985 to week 5 June 2021 (in Medline), and 7 July 2021 (in Embase and Scopus), in English or Spanish languages.”

4. Line 111 authors mentioned that the full paper would be checked if the conference abstract was found, but how did you address if no full paper was found?

Thank you. In fact, we found very few Conference abstracts in our search and that were all subsequently published in a peer-reviewed publication. However, if any additional ones had been identified, we would have contacted the main author asking if a full paper was available

5. Line192 When using an acronym (OA) for the first time, it must be spelled out in the text.

Thank you. The full-term has been inserted in the corresponding line 

6. According to “PRISMA 2020 explanation and elaboration: updated guidance and exemplars for reporting systematic reviews”, a general interpretation of the results should be provided. The first paragraph of the discussion is too detailed. A concise and informative paragraph might be more appropriate here.

Thank you. We have revised accordingly as follows:

“This systematic review examined the evidence about exposure to high-intensity or physically-demanding activities either at work or in leisure-time and the risk of hip or knee revision surgery. From 11,307 studies identified as of interest, 13 fulfilled our inclusion/exclusion criteria. Amongst five studies exploring the role of occupation (or occupational activities) after hip arthroplasty, two reported a positive association with pre-operative farming [29, 30]; one reported an increased risk with heavy physical work (pre and post-THA) [28], and one no effect [31]. The only study of occupational activities after TKA found no association [36]. For revision THA with sports and LTPA participation, 4 studies were inconsistent: two found increased risk [28, 31], (only amongst men) [28]; one found no effect [29]; and one (poor-quality) study found reduced risk [26]. Three studies evaluated THA revision and total activities (work and leisure) using the same measurement tool (UCLA) but were also conflicting: one suggested an increased risk with increasing activity [32]; one suggested an increased risk with high-impact sport [33]; and the best quality study found no effect for either level or intensity of activities [34]. Another (weak) study measured total activities with a different tool and reported a doubling of risk of revision THA [27]. For LTPA after TKA, one study found no association [36] , another found better implant survival with more physical activity [38], one found that total physical activity at the time of TKA increased risk of revision for all causes, but not after exclusion of those performed for infectious causes [37] and another (poor quality) study reported that LTPA increased the risk of revision [35]. Taken together, we found a heterogeneous literature unsuitable for pooling for quantitative synthesis. The evidence is unconvincing for an increased risk of revision after hip or knee arthroplasty associated with LTPA, and although there is some evidence for increased risk of revision THA with physically-demanding work, more research is required using standardised methodology. In particular, more studies are needed after TKA.”

Reviewer #2: 

Thank you for giving me the opportunity to review this manuscript. Well-performed systematic reviews have high clinical importance. My compliments to the authors for a well performed systematic review. 

Thank you – we appreciate the positive feedback.

My main concern is about how the discussion is lined out. Besides that, I only have a few comments.

Title

Appropriate

Abstract

I recommend not to use emphasising words like “very” in a scientific paper.

Thank you. We have removed the “very”.

Introduction

Gives a good rationale for the study.

Materials and methods

Well described.

Results

Well written.

Discussion

In the first page on the discussion, all results are repeated, without any actual discussion. I recommend the authors to start with a short summary of main findings, and then discuss the results, in relation to other studies.

Thank you for your suggestion. In light of this comment, and that of Reviewer 1, we have re-drafted

“This systematic review examined the evidence about exposure to high-intensity or physically-demanding activities either at work or in leisure-time and the risk of hip or knee revision surgery. From 11,307 studies identified as of interest, 13 fulfilled our inclusion/exclusion criteria. Amongst five studies exploring the role of occupation (or occupational activities) after hip arthroplasty, two reported a positive association with pre-operative farming [29, 30]; one reported an increased risk with heavy physical work (pre and post-THA) [28], and one no effect [31]. The only study of occupational activities after TKA found no association [36]. For revision THA with sports and LTPA participation, 4 studies were inconsistent: two found increased risk [28, 31], (only amongst men) [28]; one found no effect [29]; and one (poor-quality) study found reduced risk [26]. Three studies evaluated THA revision and total activities (work and leisure) using the same measurement tool (UCLA) but were also conflicting: one suggested an increased risk with increasing activity [32]; one suggested an increased risk with high-impact sport [33]; and the best quality study found no effect for either level or intensity of activities [34]. Another (weak) study measured total activities with a different tool and reported a doubling of risk of revision THA [27]. For LTPA after TKA, one study found no association [36] , another found better implant survival with more physical activity [38], one found that total physical activity at the time of TKA increased risk of revision for all causes, but not after exclusion of those performed for infectious causes [37] and another (poor quality) study reported that LTPA increased the risk of revision [35]. Taken together, we found a heterogeneous literature unsuitable for pooling for quantitative synthesis. The evidence is unconvincing for an increased risk of revision after hip or knee arthroplasty associated with LTPA, and although there is some evidence for increased risk of revision THA with physically-demanding work, more research is required using standardised methodology. In particular, more studies are needed after TKA.”

Page 31, line 443-454: this paragraph would be more relevant in the introduction.

Thank you. We have moved this section to the Introduction as suggested. Please see Page 4 Line 85

Page 32, line 455-463: this part does not seem to have any relation with the results from the present study.

Once again, thanks for your suggestion. We deleted this section.

Page 32, line 465-480: introduction, not discussion

This has been deleted

Conclusion

The first part in the conclusion is relevant. The last part, with suggestions for future research is not a conclusion and should be moved to discussion.

Thank you, the Conclusion has been shortened as suggested (Page 34 Line 500):

“In summary, the findings from this review highlighted the paucity of relevant studies on this research question, especially for revision surgery after TKA. Many studies only assessed relevant exposure pre-operatively, which is likely to be of limited relevance to post-operative activities. Based on the limited evidence identified, occupation and leisure-time physical activity do not convincingly increase the risk of revision after hip or knee arthroplasty. Given the lack of evidence and the inconsistencies found, more research is needed to assess the risk of mechanically loading the replaced hip or knee following joint arthroplasty, and, in particular, to investigate the impact of return to physically-demanding occupational activities, given that increasing numbers of people will want and need to return to work post-arthroplasty.” 

Tables and figures

Appropriate

References

Appropriate

REVIEWER #3:

Thank you for allowing me to review your interesting manuscript on the evidence for risk of revision arthroplasty related to physically demanding occupation and leisure time activities

Thank you for your detailed review and helpful comments.

General remark: Please carefully check the manuscript for grammatical errors and incorrect sentencing once more

Thank you for this. We have carefully revised the manuscript and re-phrased some sentences as necessary. 

Title: no comments

Abstract: introduction is not logical to me, please make it more clear to the reader how you get from primary osteoarthritis to revision arthroplasty in this intro

Apologies. In order to fit within the abstract word count, we had been too brief. This has been re-drafted to make a more logical flow. Please see Page 2 line 31 of ‘Manuscript’ document:

“However, physically-demanding activities can cause primary osteoarthritis and accordingly such exposure post-operatively might cause prosthetic failure.”

Add July to ‘2021’ for your search. 

Thank you. This has been added as recommended:

“We searched Medline, Embase and Scopus databases (1985 - July 2021) for original studies including primary lower limb arthroplasty recipients that gathered information on physically-demanding occupational and/or leisure activities and rates of revision arthroplasty.”

The conclusion of your abstract does not state a conclusion based on your results (effect of PA on revision risk). 

Thank you. We have re-drafted to better achieve this without going over the recommended word count, as follows:

Conclusion 

“There is currently a limited evidence base to address this important question. There is weak evidence that the risk of revision hip arthroplasty may be increased by exposure to physically-demanding occupational activities but insufficient evidence about the impact on knee revision and about exposure to leisure-time activities after both procedures. Considerable variability in methodological assessments in terms of activity and follow-up between studies limited interpretation of results. More evidence is urgently needed to be able to advise lower limb arthroplasty recipients about the risk of revision associated with physically-demanding activities. This is particularly, particularly important for people hoping expecting to return to jobs in some sectors (manufacturing, e.g. construction, agriculture, military, fire services, professional athletes).”

Introduction: well written

line 77: your reference only supports this statement for knee arthroplasty patients

Thank you for raising this point. The sentence has been re-worded using figures for hip and knee replacement as follows: (Page 4 Line73)

“Although highly effective interventions [7, 8], hip and knee replacements may fail over time necessitating revision surgery to the replaced joint. Revision surgery is more complex than primary arthroplasty with poorer outcomes [9] and a greater economic burden on health services [10, 11]. Survival rates after arthroplasty are lower amongst younger recipients. One studied reported higher failure rate in hip arthroplasty recipients aged <60 years [12]. Another study reported that, compared with the 15% lifetime risk of revision amongst those aged 60 years, rates of hip revision were 29.6% and of knee revision were 35.0% amongst those aged 50-54 years [13]. These age differences are at least partly explained by sex (greater risk among male recipients) but also by different indications for primary surgery, type of prosthesis and fixation method [14] but there is need for a better understanding of the impact of other factors on implant survival.” 

Material and methods:

Line 130 screening for title and abstract by one author, with only a random 10% checked by two other authors, limits the validity of this first screening step in my opinion. 

We apologise that, in the interests of brevity, we have failed to fully explain our Methods. In fact, title and abstract screening was initially undertaken by one reviewer who classified papers as “eligible”, “ineligible” or “uncertain as to whether eligible or not” (n=229). All papers in the “uncertain” category were reviewed by a second reviewer (ECH or CHL) and in the event of lack of consensus, were discussed with a third reviewer (KWB). Additionally, a random sample of 10% of the papers selected as “eligible” or “ineligible” by the first reviewer were also checked by a second reviewer and none were found to have been mis-classified. 

Please see the changes introduced to clarify this point in Page 7 line 145 of ‘Manuscript’ document:

“Screening of titles and abstracts was initially undertaken by one reviewer (EZ) who classified papers as “eligible”, “ineligible” or “uncertain whether eligible or not” for inclusion in the review. A second reviewer (ECH and CHL), checked all papers classified as uncertain to be suitable for inclusion (n=229) and where consensus was not reached, discussed with a third reviewer (KWB). Additionally a random sample of 10% of those deemed by the first reviewer as “eligible” or “ineligible” were also screened by a second reviewer but it was demonstrated that none of these papers had been misclassified. Once full text papers had been agreed and selected, two reviewers (EZ, ECH or CHL) independently reviewed the full texts for suitability for inclusion. Discrepancies were discussed by both reviewers and, if consensus was not reached, with a third reviewer (KWB).”

Information on grading of physically-demanding activities (low, intermediate, high) is lacking in the methods section. How did the authors define this?

Thank you for this important question. Ideally, to address the research question, we would have hoped to be able to “pool” data from different studies but, as the Reviewer has correctly deduced, data were presented in such different ways by different authors that we have been forced to extract from each paper exactly which tools or questionnaires had been used to collect information about exposure to occupational and/or leisure-time physical activity, as well as the methods used by the authors to grade the exposure, and reported it in the current review in the same format (for example active vs inactive). We have added more to the paragraph to clarify this: Please see the resulting paragraph below in Page 7 Line 158 of ‘Manuscript’ document:

“Data extraction included: author and year of publication, study design, country, site of procedure, duration of follow-up, indication for primary arthroplasty, sample size, age at the time of primary operation and age at revision (if provided), number lost to follow-up, operation-related factors, definition of revision, type of physical activity (undertaken at work and/or during leisure time), method of measurement of physical activity including how exposure to physical activity that loads the joint was categorised (e.g. “active vs inactive” or “high, medium, low”), covariates considered, risk estimates and source of funding where available.”

Results: 

Overall, the results section is too long. 

Given the importance of the clinical question and the heterogeneity of the retrieved literature, we felt that a detailed narrative synthesis was indicated in particular to draw out the great differences between studies in terms of a) measurement of exposure to physical activity, which translates into different ways of categorising people, and b) the frame time when the exposure was collected since this is a key factor to assess the effect of physical activity, whichever its nature, on revision surgery. If the Editor feels that additional shortening is required, we will of course seek to achieve this.

Suggest adding your quality assessment as supplementary material. Did you do anything with the quality assessment, i.e. did you sort your results based on the quality of the articles?

Thank you for making this suggestion. The quality assessment has been uploaded as Supplementary Material; S2 Table and S3 Table. 

Regarding the criteria used to describe the information/findings from the papers, we used chronological order of publication to summarise the information presented in Tables 1 to 4, whereas quality assessment was used in the narrative synthesis when describing the findings from the studies. We opted to present the narrative synthesis sorted by the timing of occurrence of the relevant exposure (pre – post or both pre and post-arthroplasty), as this is an important context within which to understand our findings. 

Line 179-180 incomplete sentence. 

Thank you for this point. It was an interesting feature of this literature review that the same paper could be scored differently for quality because of differences in the assessment methods for measurement of leisure-time, as compared with occupational, activities. We have re-written in order to make this clearer (Line199 of ‘Manuscript’ document):

“One study reported exposure to LTPA more precisely than was the case for exposure to physically-demanding occupational activities (not stated how many people actually returned to the occupations post-operatively) and therefore, according to our quality assessment criteria, needed to be scored differently for the purposes of this review [28]: it was graded acceptable quality for LTPA but poor quality for occupation.”

Line 190: 5,328 is misleading because the authors only included 1,008 in each group for their analysis. 

Once again thank you for spotting this error. The figure provided in the text corresponds to the sample the study began with. Subsequent correction with the accurate figure has been introduced in Page 15 Line 213. 

“The number of study participants ranged from 18 [33] to a maximum of 2,016 [37], and the post-operative follow-up from 4.9 [27] to 11 years [33, 38].” 

Line 193: I’m surprised by the prevalence of >60%, I was under the impression that you only included primary OA cases? Only two studies recruited participants who underwent TKA for primary OA. 

Thank you for this other important point. Ideally, we would have focused the review only to papers which investigated this question in relation to arthroplasty undertaken for a single indication but we found that so few studies assessed exposure to work and non-work-related physical activity in relation to lower limb revision surgery that we needed to be inclusive. Only 2 out of the 13 eligible papers selected only patients in whom primary OA was the indication for lower limb arthroplasty (Jones et al and Ponzio et al). In the remaining 11 papers, primary OA was the most common indication for the primary operation, but participants with secondary OA as the indication were also included (authors did not present their results separately by diagnosis). 

Line 266-268: discussion remark

Thank you. The sentence has been removed. 

Line 283-285: discussion remark

Thank you. The sentence has been removed. 

Line 315-316: discussion remark

Thank you. The sentence has been removed.

Table 4: last column has ‘Risk estimate 95% CI’ as a header but most studies did not report a 95% CI so this header is not accurate here.

Thank you, Table 4 includes four studies that compared people exposed to different levels of physical activity. The header of the column has been slightly modified to cover not only the risk ratios but also the p-values provided. Please see Table 4 in Page 25.

Line 333: did the authors also combine all reasons for revision in the low- and high-activity cohorts? Would be more interesting than the separate analyses for each reason for revision.

The Reviewer is correct - the authors also compared prevalence rate of revision for all causes between the active and inactive group. We did not present this information in the previous draft of the paper because revision for “all causes” includes infection and we have some difficulty with the biological relevancy of physical activities to risk of infection. In light of the Reviewer’s question however, we have added this information to Table 4. We have also added this to the narrative review, Page 28 Line 351 of ‘Manuscript’ document, as shown below:

“At 5 to 10 years’ post-operation, the revision rate for all causes (including infection) was different between the active and inactive groups (p=0.019), whereas revision rate for all non-infective causes was not statistically significantly different between active and inactive groups.”

Discussion:

Could be shorter but I enjoyed reading it.

Thank you for your kind comment. 

Line 378 and throughout the manuscript ‘as to whether’ is used excessively.

Four of the five uses of “as to whether” have been re-phrased, leaving just one in this version.

Line 382: these numbers don’t necessarily suggest that the topic is understudied; maybe the 13 existing studies sufficiently answered the question? 

We acknowledge the Reviewer’s point. If we had retrieved 13 large, high-quality studies we might have deemed that this topic was “well-studied”. However, the fact that only 13 papers our of >11,000 covered the topic, and they did so differently and rather poorly, led us to make this statement. Based on the comment however, we have removed this phrase. In particular, we were disappointed that only three papers have investigated this question after knee arthroplasty.

Conclusion:

Please provide the reader with a general summary of your own results, not just the statement that limited evidence is available (they know by now and this is not the most important finding you want them to remember, I guess?). “In conclusion, based on limited evidence, occupational PA and LTPA do not convincingly increase the risk of revision hip and knee arthroplasty in primary arthroplasty patients” or something like that.

Thank you for the suggestion to improve the conclusion of the manuscript. We have introduced a sentence summarising what we found in this systematic review (Page 34 Line 501)

“Many studies only assessed relevant exposure pre-operatively, which is likely to be of limited relevance to post-operative activities. Based on the limited evidence identified, occupation and leisure-time physical activity do not convincingly increase the risk of revision after hip or knee arthroplasty.”

References

No comments

Editor comments

Please ensure that your manuscript meets PLOS ONE's style requirements, including those for file naming. The PLOS ONE style templates can be found at:

Of course, we have carefully checked that the manuscript complies with the journal style requirements.

Please provide a table reporting in detail the results of your quality assessment, showing how each included study scored on every item of the modified Scottish Intercollegiate Guidelines Network (SIGN) checklist and Assessment of Quality in Lower Limb Arthroplasty checklist.

Thank you for your comment. As mentioned above, in the responses to Reviewer 3, we have added two Supplementary Tables showing the results of the quality assessment: one for cohort studies (S2 Table) and another one for case-control studies (S3 Table).

In your Data Availability statement, you have not specified where the minimal data set underlying the results described in your manuscript can be found. PLOS defines a study's minimal data set as the underlying data used to reach the conclusions drawn in the manuscript and any additional data required to replicate the reported study findings in their entirety. All PLOS journals require that the minimal data set be made fully available. For more information about our data policy, please see http://journals.plos.org/plosone/s/data-availability.

We thank you the editor for raising this point. This is systematic review therefore all relevant data are within the manuscript and the information can be accessed downloading the studies identified in the search.

The cover letter specifies an update for the Data Availability statement.

We note that you have stated that you will provide repository information for your data at acceptance. Should your manuscript be accepted for publication, we will hold it until you provide the relevant accession numbers or DOIs necessary to access your data. If you wish to make changes to your Data Availability statement, please describe these changes in your cover letter and we will update your Data Availability statement to reflect the information you provide.

Once again, thank you for spotting the mistake. We understand that the response to the previous comment clarifies that the are no data to upload in the repository.

---

## [Decision Letter · Decision Letter 1]

14 Feb 2022

Risk of revision arthroplasty surgery after exposure to physically demanding occupational or leisure activities: a systematic review

PONE-D-21-27689R1

Dear Dr. Zaballa Lasala,

We’re pleased to inform you that your manuscript has been judged scientifically suitable for publication and will be formally accepted for publication once it meets all outstanding technical requirements.

Kind regards,

John Leicester Williams, Ph.D.

Academic Editor

PLOS ONE

Additional Editor Comments (optional):

Reviewers' comments:

Reviewer's Responses to Questions

**Comments to the Author**

1. If the authors have adequately addressed your comments raised in a previous round of review and you feel that this manuscript is now acceptable for publication, you may indicate that here to bypass the “Comments to the Author” section, enter your conflict of interest statement in the “Confidential to Editor” section, and submit your "Accept" recommendation.

Reviewer #1: All comments have been addressed

Reviewer #2: All comments have been addressed

Reviewer #3: All comments have been addressed

2. Is the manuscript technically sound, and do the data support the conclusions?

Reviewer #1: Yes

Reviewer #2: Yes

Reviewer #3: Yes

3. Has the statistical analysis been performed appropriately and rigorously? 

Reviewer #1: Yes

Reviewer #2: Yes

Reviewer #3: Yes

4. Have the authors made all data underlying the findings in their manuscript fully available?

Reviewer #1: Yes

Reviewer #2: Yes

Reviewer #3: Yes

5. Is the manuscript presented in an intelligible fashion and written in standard English?

Reviewer #1: Yes

Reviewer #2: Yes

Reviewer #3: Yes

6. Review Comments to the Author

Reviewer #1: All of my questions have been addressed very well and in detail. I have no further questions or concerns.

Reviewer #2: Thank you for your revised manuscript, which is much improved. Congratulations to a well performed study!

Reviewer #3: No more comments, the authors have made a great effort in improving the manuscript and apart from some spelling issues I think it is now ready for publication.

7. PLOS authors have the option to publish the peer review history of their article (what does this mean?). If published, this will include your full peer review and any attached files.

Reviewer #1: No

Reviewer #2: No

Reviewer #3: **Yes: **Alexander Hoorntje

---

## [Editor Report · Acceptance letter]

18 Feb 2022

PONE-D-21-27689R1 

Risk of revision arthroplasty surgery after exposure to physically demanding occupational or leisure activities: a systematic review 

Dear Dr. Zaballa :

I'm pleased to inform you that your manuscript has been deemed suitable for publication in PLOS ONE. Congratulations! Your manuscript is now with our production department. 

Kind regards, 

on behalf of

Dr. John Leicester Williams 

Academic Editor

PLOS ONE